# Leveraging automatic strategy discovery to teach people how to select better projects

## Abstract

Human decisions are often suboptimal due to limited cognitive resources and time constraints. Prior work has shown that errors in human decision-making can in part be avoided by leveraging artificial intelligence to automatically discover efficient decision strategies and teach them to people. So far, this line of research has been limited to simplified decision problems that are not directly related to the problems people face in the real world. Current methods are mainly limited by the computational difficulties of deriving efficient decision strategies for complex real-world problems through metareasoning. To bridge this gap, we model a real-world decision problem in which people have to choose which project to pursue, and develop a metareasoning method that enables us to discover and teach efficient decision strategies in this setting. Our main contributions are: formulating the metareasoning problem of deciding how to select a project, developing a metareasoning method that can automatically discover near-optimal project selection strategies, and developing an intelligent tutor that teaches people the discovered strategies. We test our strategy discovery method on a computational benchmark and experimentally evaluate its utility for improving human decision-making. In the benchmark, we demonstrate that our method outperforms PO-UCT while also being more computationally efficient. In the experiment, we taught the discovered planning strategies to people using an intelligent tutor. People who were trained by our tutor showed a significant improvement in their decision strategies compared to people who tried to discover good decision strategies on their own or practiced with an equivalent tutor that did not reveal the optimal strategy. Project selection is a very consequential high-stakes decision regularly faced by organizations, companies, and individuals. Our results indicate that our method can successfully improve human decision-making in naturalistic settings similar to the project selection decisions people face in the real-world. This is a first step towards applying strategy discovery methods to improve people's decisions in the real-world.

## 1  Introduction

Corporations and individuals commonly have to select a project to pursue out of multiple alternatives. These project selection problems are usually high-stakes decisions that can be highly impactful for the future of an organization. For example, an organization looking for a sustainable investment project [17] could benefit both financially and by improving its public image by selecting an impactful and profitable project, or incur major losses by selecting an unsuitable project.

Previous research on project selection recommends that candidate projects should be evaluated by multiple experts [9, 17, 21], and many structured approaches to integrate the experts' opinions exist [11]. However, structured project selection techniques are not well utilized in the real-world [28, 21],

and decision-makers often rely on their intuition and much simpler techniques like brainstorming [18]. This is concerning because the intuitive decisions of groups and individuals are highly susceptible to biases and unsystematic error [16]. However, people's errors in decision-making can partly be prevented by teaching them better decision strategies. This approach is known as *boosting* [15].

To identify appropriate decision strategies, we can score candidate strategies by their *resource-rationality*, that is the degree to which they make rational use of people's limited time and bounded cognitive resources [19]. In the resource-rational framework, the decision operations people can perform to arrive at a decision are modeled explicitly and assigned a cost. The overall efficiency of a decision strategy $h$ in an environment $e$ can then be computed by subtracting the expected costs $\lambda$ of the $N$ used decision operations from the expected utility $R_{total}$ of the resulting decision (see Equation 1) [10]. This measure is called resource-rationality score (*RR-score*) [10]. People are usually not fully resource-rational, and identifying decision strategies would enable people to perform as well as possible is an important open problem [6, 10, 14, 22].

$$\mathrm{RR}(h, e) = \mathbb{E}[R_{total}|h, e] - \lambda\mathbb{E}[N|h, e] \tag{1}$$

Recent work has demonstrated that the theory of resource rationality makes it possible to leverage AI to automatically discover and teach decision strategies that enable real people to make their decisions as well as possible [6, 10, 1, 32, 22]. This approach has been dubbed *AI-powered boosting*. The first step of AI-powered boosting is to compute resource-rational decision strategies. Automatic strategy discovery methods [6, 10, 14, 32, 22] can discover efficient decision strategies by solving the metareasoning problem of deciding which decision operations to perform. While recent work has extended automatic strategy discovery methods to larger [10] and partially observable environments [14], so far, they have not been applied to real-world decisions such as project selection.

In this article, we extend AI-powered boosting to improve how people select projects. Project selection is challenging because many crucial attributes of the candidate projects, such as their expected profitability, cannot be observed directly. Instead, they have to be inferred from observations that are not fully reliable. We, therefore, formalize project selection strategies as policies for solving a particular class of partially observable Markov Decision Processes (POMDPs). This formulation allows us to develop the first algorithm for discovering resource-rational strategies for human project selection.To achieve this, we model a realistic project selection task as a metareasoning problem. The metareasoning consists in deciding which information one should request from which advisors when information is highly limited, uncertain, and costly. We develop an efficient algorithm for solving this problem and apply it to derive the optimal decision strategy for a project selection problem a financial institution faced in the real world [17]. Finally, we develop an intelligent tutor [6] that teaches the decision strategy discovered by our method to people. We evaluated our approach by letting our intelligent tutor teach the automatically discovered project selection strategy to about 100 people, and then evaluated the quality of their decisions in realistic project selection problems against two control groups. Our results indicate that our approach can successfully improve human decisions in real-world problems where people are reluctant to let machines decide for them.

## 2 Background

**Project selection**   In the project selection problem, a decision-maker aims to select the best-fitting project out of several candidates [27]. Apart from a project's profitability, the evaluation usually also considers other factors, such as the alignment with organizational goals [7]. This problem can be formalized as multi-criteria decision-making (MCDM) [11, 23]. Projects can be evaluated using a scoring technique, which evaluates relevant criteria and then combines them to a weighted sum [27]. Common approaches to solving the project selection problem include techniques such as the analytic hierarchy process, the analytic network process, real options analysis, and TOPSIS (see [11] for a review). These methods are commonly combined with fuzzy sets to account for uncertainty [17]. However, these methods are rarely used in real-world problems because implementing them would require gathering and integrating a lot of information through a time-consuming process, which is often incompatible with the organizational decision process [28, 21]. Instead, organizations often rely on simpler, less structured methods like brainstorming [18]. In addition, the detailed information required by these methods can be costly to acquire in real-world settings.

**Judge-advisor systems**  In a Judge Advisor System (JAS) [2], typically, a single decision-maker has to make a decision, and multiple advisors support the decision-maker by offering advice. Variations of the task can include costly advice [33, 12] , or advisors with varying reliability [24]. This is a common situation when CEOs decide which project their company should launch next. Unfortunately, decision-makers are known to be highly susceptible to systematic errors, such as weighing one's own opinion too strongly, overconfidence, egocentric advice discounting, and weighting the recommendations of advisors by their confidence rather than their competence [2, 25, 33].

**Strategy discovery methods**  Discovering resource-rational planning strategies can be achieved by solving a meta-level Markov decision process [13, 4, 5], which models the metareasoning process as a Markov Decision Process (MDP), which state represents the current belief about the environment and actions represent decision operations. Performing a decision operation results in a negative cost and results in an update to the belief state. A special termination action represents exiting the metareasoning process and making a real-world decision, guided by the current beliefs [13]. Multiple methods for solving meta-level MDPs exist (e.g. [26, 4, 13, 10]). We refer to these algorithms as strategy discovery methods [4, 6, 32, 10, 14, 22]. They learn or compute policies for selecting sequences of cognitive operations (i.e., computations) people can perform to reach good decisions.

MGPO [14] is currently the only strategy discovery algorithm that can efficiently approximate resource-rational strategies for decision-making in partially observable environments. MGPO chooses decision operations by approximating their value of computation [26] in a myopic manner: it always selects the computation whose execution would yield the highest expected gain in reward if a decision had to be made immediately afterward, without any additional planning.

**Cognitive tutors**  Past work has developed cognitive tutors that teach automatically discovered planning strategies to people [6, 32, 22, 10]. Training experiments indicated that training with these cognitive tutors could significantly boost the quality of people's planning and decision-making [6, 32, 10, 14, 22]. These cognitive tutors teach efficient decision strategies in an automated manner, usually by computing the value of available decision operations using strategy discovery methods, and providing the learner feedback on the quality of the computations they select. Initially limited to small planning tasks due to the computational complexity of solving meta-level MDPs [20, 6], recent work has extended existing methods to larger [10] and partially observable [14] settings. However, none of these methods have been applied to naturalistic problems so far.

A crucial obstacle is that these methods are limited to settings where all decision-relevant information comes from the same source. By contrast, in the real world, people have to choose between and integrate multiple different sources of information. In doing so, they have to take into account that some information sources are more reliable than others. Additionally, current strategy discovery methods are limited to artificial settings where each piece of information is an estimate of a potential future reward. By contrast, in the real world, most information is only indirectly related to future rewards, and different pieces of information have different units (e.g., temperature vs. travel time).

## 3  Formalizing optimal decision strategies for human project selection as the solution to a meta-level MDP

In this section, we introduce explain our general resource-rational model of project selection, which we expect to be widely applicable to concrete, real-world project selection problems.

Our model of project selection consists of two decision problems, an object-level decision-problem and a meta-level MDP [4, 13]. The two decision problems separate the actions the decision-maker has to choose between (object level), such as executing one project versus another, from decision operations that represent thinking about which of those object-level actions to perform (meta-level), such as gathering information about the projects' attributes. This allows us to solve both problems separately. The object-level decision problem is a MCDM problem, where a set of $N_P$ potential projects $\mathcal{P} = \{p_1, ... p_{N_P}\}$ are evaluated using $N_C$ relevant criteria $C = [c_1, ... c_{N_C}]$ weighted by fixed predetermined weights $W = [w_1, ... w_{N_C}]$. Actions in the object-level problem represent selecting the corresponding project ($\mathcal{A} = \{a_1, ..., a_{N_P}\}$). The reward of a selecting a project is computed by summing the weighted criteria scores of the selected project: $r^O(a_i) = \sum_c w_c c_{c,i}$ [9].

While the object-level decision problem is our model of the project selection task, the meta-level MDP is our formalization of the problem of discovering resource-rational project selection strategies. It models the task of gathering information about deciding which project to select. States in the meta-level MDP are belief states that represent the current information about each project's attributes. We model belief states using a multivariate Normal distribution to quantify the estimated value and uncertainty about the $N_P$ projects' scores on the $N_C$ criteria: $b = [(\mu_{1,1}, \sigma_{1,1}), \cdots, (\mu_{N_P,N_C}, \sigma_{N_P,N_C})]$. The actions (decision operations) of the meta-level MDP gather information about the different attributes of projects by asking one of the $N_E$ experts for their estimate of how one of the projects scores on one of the criteria. Experts provide discrete estimates from $min_{obs}$ to $max_{obs}$, and expert estimates can differ in their reliability and their cost. Specifically, the available actions are $A^M = \{a_{1,1,1}, \cdots, a_{N_P,N_C,N_E}, \perp\}$, where the meta-level action $a_{i,j,k}$ represents asking the expert $e_k$ for their estimate of criterion $c_j$ of project $p_i$. After receiving information $obs$ from an expert, the current belief state $b_{p_i c_j} = \mathcal{N}(\mu, \sigma)$ is updated by applying the update equation for a Gaussian likelihood function with standard deviation $\sigma_e$ (i.e. the expert's reliability) and a conjugate Gaussian prior (i.e., the current belief), that is $\hat{\mu} \leftarrow \left( \frac{w_{c_i} \cdot \mu}{(w_{c_i} \cdot \sigma)^2} + \frac{w_{c_i} \cdot obs}{(w_{c_i} \cdot \sigma_e)^2} \right) \cdot \left( (w_{c_i} \cdot \sigma)^2 + (w_{c_i} \cdot \sigma_e)^2 \right)$ and

$\hat{\sigma} \leftarrow \sqrt{\frac{1}{\frac{1}{(\sigma \cdot s)^2} + \frac{1}{(\sigma_e \cdot s)^2}}}$.

The reward of these meta-level actions is the negative cost $r^M(a_{i,j,k}) = -\lambda_{e_k}$ of asking the expert $e_k$ for help. Finally, the meta-level action $\perp$ is the termination action, which corresponds to terminating the decision-making process and selecting a project. The reward of the termination action is the expected reward of selecting the best project according to the current belief-state. An optional budget parameter $N_a$ specifies the maximum number of available meta-level actions, after which the termination action is performed automatically.

Meta-level MDPs are notoriously difficult to solve due to their extremely large state space [10, 13]. In the project selection task, the meta-level MDP has $(max_{obs} - min_{obs} + 2)^{N_P \cdot N_C \cdot N_E}$ possible belief states and $N_P \cdot N_C \cdot N_E + 1$ possible meta-level actions. Our meta-level MDP introduces multiple new intricacies that current metareasoning methods like MGPO [14] aren't equipped to handle, making strategy discovery in this setting especially difficult. Compared to previously formulated meta-level MDPs [4, 13, 10, 14, 22], our meta-level description of project selection differs in the following ways: (1) the maximum amount of meta-level actions is constrained with a budget, (2) the project selection task features multiple consultants who differ in both the quality of their advice and the cost of their services, (3) consultants in the project selection task offer discrete estimates of each criterion , requiring that (4) criteria ratings are scaled to allow weighting the criteria according to their importance.

# 4 A new metareasoning algorithm for discovering optimal decision strategies for human project selection

Previous metareasoning methods are unable to handle some of the intricacies of the project selection problem. Therefore, we developed a new strategy discovery method based on MGPO [14], which overcomes the limitations that prevent MGPO from being applicable to project selection. To reflect the commonalities and innovations, we call our new strategy discovery method MGPS (meta-greedy policy for project selection). Similar to MGPO, our method approximates the value of computation (VOC) [26] by myopically estimating the immediate improvement in decision quality. Improving upon MGPO, MGPS calculates the myopic approximation to the VOC in a way that accounts for discrete criteria ratings, criteria scaling, and multiple sources of information with different costs and reliabilities.

MGPS calculates a myopic approximation to the VOC of asking an expert about a specific criterion of a single project according to Algorithm 1. To account for discrete advisor outputs, Algorithm 1 iterates over the discrete set of possible ratings the expert might give and estimates the probability $p_{obs}$ of each rating ($obs$) and the belief update that would result from it $\hat{\mu}_{obs}$ . The probability of each rating is computed using the cumulative distribution function ($\Phi$) of the belief state for the selected project's criterion score (see Line 7). Here, the standard deviation $\sigma_e$ of the likelihood function encodes the expert's reliability, the prior ($\mathcal{N}(\mu, \sigma)$) is the current belief about the project's score on the evaluated criterion, and the weights $w_{c_i}$ convert the criteria into a common currency. The belief update that would result from the observation ($\hat{\mu}_{obs}$) is computed by applying the belief state update

---

**Algorithm 1** MGPS VOC calculation for an action $a_{p_i, c_i, e_i}$

---

1: **function** MYOPIC_VOC($p_i$, $c_i$, $e_i$, $b$)
2:   $r_p \leftarrow \mathbb{E}[r^O(p_i)]$
3:   $r_{alt} \leftarrow \max_{p_j \in \mathcal{P} - \{p_i\}} \mathbb{E}[r^O(p_j)]$
4:   $\mu, \sigma \leftarrow b_{p_i c_i}$
5:   **for** $obs$ **from** $min_{obs}$ **to** $max_{obs}$ **do**
6:    **if** $min_{obs} < obs < max_{obs}$ **then**
7:     $p_{obs} \leftarrow \Phi\left(\frac{w_{c_i} \cdot (obs + 0.5 - \mu)}{\sqrt{(w_{c_i} \cdot \sigma)^2 + (w_{c_i} \cdot \sigma_e)^2}}\right) - \Phi\left(\frac{w_{c_i} \cdot (obs - 0.5 - \mu)}{\sqrt{(w_{c_i} \cdot \sigma)^2 + (w_{c_i} \cdot \sigma_e)^2}}\right)$
8:    **else if** $obs == min_{obs}$ **then**
9:     $p_{obs} \leftarrow \Phi\left(\frac{w_{c_i} \cdot (obs + 0.5 - \mu)}{\sqrt{(w_{c_i} \cdot \sigma)^2 + (w_{c_i} \cdot \sigma_e)^2}}\right)$
10:    **else**
11:     $p_{obs} \leftarrow 1 - \Phi\left(\frac{w_{c_i} \cdot (obs - 0.5 - \mu)}{\sqrt{(w_{c_i} \cdot \sigma)^2 + (w_{c_i} \cdot \sigma_e)^2}}\right)$
12:    **end if**
13:    $\hat{\mu}_{obs} \leftarrow \left(\frac{w_{c_i} \cdot \mu}{(w_{c_i} \cdot \sigma)^2} + \frac{w_{c_i} \cdot obs}{(w_{c_i} \cdot \sigma_e)^2}\right) \cdot \left((w_{c_i} \cdot \sigma)^2 + (w_{c_i} \cdot \sigma_e)^2\right)$
14:   **end for**
15:   **if** $r_p > r_{alt}$ **then**
16:    voc $\leftarrow \sum_{obs=min_{obs}}^{max_{obs}} p_{obs}(r_{p_{alt}} + \mu - r_p - \hat{\mu}_{obs}) \cdot \mathbb{1}(r_p - \mu + \hat{\mu}_{obs} < r_{alt})$
17:   **else**
18:    voc $\leftarrow \sum_{obs=min_{obs}}^{max_{obs}} p_{obs}(r_p + \hat{\mu}_{obs} - \mu - r_{p_{alt}}) \cdot \mathbb{1}(r_p - \mu + \hat{\mu}_{obs} > r_{alt})$
19:   **end if**
20:   **return** $(1 - w_\lambda)$voc - $w_\lambda \lambda_{e_i}$
21: **end function**

---

(see Line 13 and Equation 3). For the highest and lowest possible ratings, we instead calculate $p_{obs}$ using the open interval (see Lines 9 and 11). The updated expected value of the belief state according to an observation $obs$ is then calculated using Bayesian inference to integrate the new observation into the belief state (see Line 13).

The VOC calculation depends on the posterior belief states that would result from the different possible observations ($\hat{\mu}_{obs}$), weighted by their probabilities. If the evaluated project currently has the highest expected reward (see Line 15), the VOC calculation expresses the probability of observing a value low enough that the second-best project becomes the most promising option (see Line 16). In the case where the evaluated project did not have the highest expected termination reward, the VOC calculation expresses the probability of observing a value high enough to make the evaluated project the most promising option (see Line 18). Finally, the cost of the requested expert $\lambda_{e_i}$ is weighted using a free cost weight parameter $w_\lambda$ and subtracted from the VOC estimate (see Line 20).

The full meta-greedy policy consists of calculating the VOC for all possible meta-level actions and iteratively selecting the meta-level action with the highest VOC. If no action has a positive VOC, the termination action $\perp$ is chosen.

# 5 Improving human project selection

Having developed a general metareasoning method for discovering resource-rational strategies for human project selection, we now extend it to an intelligent cognitive tutor for teaching people how to select better projects. Our goal is to provide a proof of concept for a general AI-powered boosting approach that can be used to improve human decision-making across a wide range of project selection problems. We first introduce a general approach for teaching people the project selection strategies discovered by MGPS, and then apply it to a real-world project selection problem.

## 5.1 MGPS Tutor: An intelligent tutor for teaching people how to select better projects

Our intelligent tutor for project selection (*MGPS Tutor*) trains people to select the near-optimal decision operations identified by MGPS. To achieve this, it lets people practice on a series of project

| | Project 1 | | | | | | | Project 2 | | | | | | |
| --- | --- | --- | --- | --- | --- | --- | --- | --- | --- | --- | --- | --- | --- | --- |
| | Expert 1 | Expert 2 | Expert 3 | Expert 4 | Expert 5 | Expert 6 | Current Estimate | Expert 1 | Expert 2 | Expert 3 | Expert 4 | Expert 5 | Expert 6 | Current Estimate |
| Economic effects scale: 0.02 | | | | | | | 3.6 ★★ | | | | | | | 3.6 ★★ |
| Social effects scale: 0.07 | | | | | | | 3.17 ★★ | | | | | | | 3.17 ★★ |
| Environmental effects scale: 0.22 | | | | | | | 3.6 ★★ | | | | | | | 3.6 ★★ |
| Strategic alliance scale: 0.11 | | | | | | | 3.13 ★★ | | | | | | | 3.13 ★★ |
| Organizational readiness scale: 0.47 | | | 5 | 1 | | | 3.25 ★★★ | | | | | | | 3.67 ★★ |
| Risk of investment scale: 0.12 | | | | | | | 2.3 ★★★ | | | | | | | 2.3 ★★★ |
| Estimated performance | | | | | | | 3.21 ★★★ | | | | | | | 3.4 ★★★ |

Figure 1: Example of the MGPS tutor offering a choice between requesting information from three different experts (highlighted in orange) in the simplified training task of deciding between two project alternatives. Refer to the supplemental material for an explanation of the experiment interface.

selection problems and gives them feedback. MPGS Tutor leverages MPGS in two ways: i) to pedagogically construct the set of queries the learner is asked to choose from, and ii) to give the learner feedback on their chosen query.

Building on the choice tutor by [14], our tutor repeatedly asks the learner to choose from a pedagogically chosen set of decision operations (see Figure 1) that always includes the query that MGPS would have performed. Moreover, it leverages MGPS's VOC calculation (Algorithm 1) to score the chosen query, and then provides binary feedback on its quality. If learners select a suboptimal expert, project, or criterion, they receive feedback indicating the correct choice and have to wait for a short time. The unpleasantness of having to wait serves as a penalty [6]. Otherwise, they receive positive reinforcement and the next choice is displayed. To receive positive reinforcement, the learner must select a query whose VOC is sufficiently close, as determined by a tolerance parameter $t$, to the VOC of the optimal action. We set the tolerance to $t = 0.001$.

Our tutor teaches the strategy discovered by MGPS using a novel sophisticated training schedule, which fosters learning by incrementally increasing the complexity of the training task. This learning methodology is also known as shaping [31], and has been successfully applied to teach decision strategies to humans [14]. Our training schedule varies the numbers of projects, how many different expert assessments learners have to choose between, and the specific types of expert assessments offered as choices. In total, our tutor teaches the discovered project selection strategy using ten training trials. The first seven training trials use a smaller version of the project selection task with only two projects, while the last three trials use the full environment with five projects. The number of choices gradually increases throughout training from 1 in the first training trial to 9 in the last three training trials. The tutor varies the types of choices across trials. After an initial trial with only a single choice, the tutor offers choices that focus on different criteria within the same project for two trials. Then, the tutor offers choices that focus on different experts within the same project for two trials. The remaining trials combine both types of highlights while sometimes also featuring queries about different projects and also increasing the overall number of choices.

## 5.2 Evaluating the effectiveness of MGPS Tutor in a training experiment

To evaluate if AI-powered boosting can improve human project selection, we tested the MPGS tutor in a training experiment. We tested if people trained by MPGS tutor learn more resource-rational project selection strategies. To make our assessment task as naturalistic as possible, we modelled it on a real project selection problem that was faced by an Iranian financial institution [17]. We will first describe how we modeled this real-world problem, and then the training experiment.

Table 1: Results of the human training experiment. Per condition, the normalized mean resource-rationality score and the mean click agreement are reported. For both measures, we also report the 95% confidence interval under the Gaussian assumption ($\pm 1.96$ standard errors).

| Condition | RR-score | Click Agreement |
|---|---|---|
| **MGPS Tutor** | $0.3256 \pm 0.0609$ | $0.4271 \pm 0.0201$ |
| No tutor | $-0.0227 \pm 0.0622$ | $0.2521 \pm 0.0171$ |
| Dummy tutor | $0.0225 \pm 0.0612$ | $0.2664 \pm 0.0159$ |

**A project-selection problem from the real world**   Khalili-Damghani and Sadi-Nezhad [17] worked on the real-world problem of helping a financial institution select between five potential projects with an eye to sustainability. Each project was evaluated by six advisors, who assigned scores from one to five on six different criteria. For our model of the task, we use the same number of experts, criteria, and projects, and the same criteria weights as the financial institution. The remaining parameters of the meta-level MDP were estimated as follows. We initialized the beliefs about the project's attributes by calculating the mean and the standard deviation of all expert ratings for each criterion according to [17]. We estimated the reliability of each expert by calculating the standard deviation from the average distance of their ratings from the average rating of all other experts, weighted by the number of occurrences of each guess. We estimated the cost parameter $\lambda$ of the meta-level MDP by $\lambda = \frac{cost}{stakes} \cdot r(\bot)$ to align the meta-level MDP's cost-reward ratio to its real-world equivalent. Using the expected termination reward of the environment $r(\bot) = 3.4$ and rough estimates for the stakes stakes $= \$10000000$ and expert costs $cost = \$5000$, this led to $\lambda = 0.002$. While [17] assumed all expert ratings are available for free, this is rarely the case. To make our test case more representative, we assumed that advisor evaluations are available on-request for a consulting fee. To capture that real-world decisions often have deadlines that limit how much information can be gathered, we set the maximum number of sequentially requested expert consultations to 5.

**Methods of the experiment**   We recruited 301 participants for an online training experiment on Prolific. The average participant age was 29 years, and 148 participants identified as female. Participants were paid £3.50 for completing the experiment, plus an average bonus of £0.50. The median duration of the experiment was 22 minutes, resulting in a median pay of £10.9 per hour. Our experiment was preregistered on AsPredicted and approved by the ethics commission of [removed] under IRB protocol [removed].

Each participant was randomly assigned to one of three conditions: (1) the *No tutor* condition, in which participants did not receive any feedback and were free to discover efficient strategies on their own; (2) the *MGPS tutor* condition, in which participants practiced with our cognitive tutor that provided feedback on the resource-rationality score MGPS assigns to the selected planning actions; and (3) the *Dummy tutor* condition, an additional control condition in which participants practiced with a dummy tutor comparable to the MGPS tutor, albeit with randomized feedback on which planning actions are correct. All participants practiced their planning strategy in 10 training trials and were then evaluated across 10 test trials.

We evaluated the participants' performance using two measures: their *RR-score* and *click agreement*. *RR-score's* are normalized by subtracting the average reward of a random baseline algorithm and dividing by the participant scores' standard deviation. The random baseline algorithm is defined as the policy that chooses meta-level actions at random until the maximum number of decision operations is reached. *Click agreement* measures, how well participants learned to follow the near-optimal strategy discovered by our method. Specifically, we computed for each participant, which proportion of their information requests matched the action taken by the approximately resource-rational strategy discovered by MGPS.

**Experiment results**   Table 1 shows the results of the experiment. To determine whether the condition of participants had a significant effect on the RR-score and click agreement, we used an ANOVA analysis with Box approximation [3]. The ANOVA revealed a significant effect of condition on both RR-score ($F(1.99, 293.57) = 10.48, p < .0001$) and click agreement ($F(1.99, 291.48) = 15.5, p < .0001$). We further compared the performance of participants in the *MGPS tutor* condition to participants in the two control conditions with post hoc ANOVA-type statistics and used Cohen's d

Table 2: Results of the performance evaluation. For each algorithm, we report the averge normalized resource-rationality score (*RR-Scores*) and the runtime per decision problem. For both measures, we also report the 95% confidence interval under the Gaussian assumption ($\pm 1.96$ standard errors).

| Algorithm | RR-score | Runtime (s) |
|---|---|---|
| **MGPS** | $0.9942 \pm 0.0234$ | $0.9079 \pm 0.0052$ |
| PO-UCT (10 steps) | $-0.4344 \pm 0.0106$ | $0.0175 \pm 0.0004$ |
| PO-UCT (100 steps) | $0.7309 \pm 0.0302$ | $0.1972 \pm 0.0008$ |
| PO-UCT (1000 steps) | $0.8681 \pm 0.0256$ | $2.3567 \pm 0.0028$ |
| PO-UCT (5000 steps) | $0.9054 \pm 0.0232$ | $10.8913 \pm 0.0173$ |

[8] to evaluate the size of the effects. The post hoc tests revealed that participants in the *MGPS tutor* condition achieved a significantly higher RR-score than participants in the *No tutor* ($F(1) = 17.88$, $p < .0001$, $d = .35$) and *Dummy tutor* ($F(1) = 13.4$, $p = .0002$, $d = .31$) conditions. Additionally, participants in the *MGPS tutor* reached a higher click agreement with our pre-computed near optimal strategy than participants in the *No tutor* ($F(1) = 25.08$, $p < .0001$, $d = .58$) and *Dummy tutor* ($F(1) = 19.3$, $p < .0001$, $d = .56$) conditions.

When evaluated on the same test trials and normalizing against the baseline reward and the standard deviation of the experiment results, MGPS achieves a mean reward of 1.17, demonstrating that MGPS discovers more resource-rational strategies than participants across all conditions. Although participants in the *MGPS tutor* condition performed significantly the better than participants in the other conditions, they did not learn to follow the strategy taught by the tutor exactly. Participants in the other conditions only discovered strategies with a similar *RR-score* to the random baseline strategy, with participants in the *No tutor* condition performing even worse than the random baseline strategy, and participants in the *Dummy tutor* condition outperforming the random baseline only by a small margin.

## 6 Performance evaluation

The results reported in the previous section show that MPGS can discover project selection strategies that are more effective than the strategies people discover on their own. But how does its performance compare to other strategy discovery algorithms? To answer this question, we evaluated MGPS on a computational benchmark. We chose PO-UCT [29] for comparisons because it is an established baseline for metareasoning algorithms in partially observable environments [14] and the more specialized MGPO algorithm is not applicable to project selection. PO-UCT utilizes Monte Carlo tree search to simulate the effects of different actions, resulting in more accurate results with increased computation time, making it a useful baseline for MGPS's computational efficiency and performance.

**Method**  We evaluated the effectiveness of our method in the project selection task by comparing it against PO-UCT [29] with different numbers of steps. All methods were evaluated across the same 5000 randomly generated instances of the project selection environment.

Our main performance measure was the resource-rationality score (*RR-Score* defined in Equation 1). To highlight the achieved improvements over a baseline algorithm that performs random meta-level actions, we normalized the reported *RR-scores*. Specifically, we applied a z-score transformation, subtracting the average reward of the random baseline algorithm (see Section 5.2) from the *RR-Scores* and dividing by the evaluated algorithm's *RR-Scores*' standard deviation. We analyze the differences in *RR-Scores* with an ANOVA and evaluate the size of statistical effects with Cohen's d [8]. Additionally, we compare the computational efficiency of the different methods, which is crucial for being able to provide real-time feedback in our cognitive tutor.

**Results**  As shown in Table 2, MGPS outperformed all tested versions of PO-UCT and the random baseline strategy. [1] An ANOVA revealed significant differences in the *RR-scores* of the strategies discovered by the different methods ($F(4, 24995) = 2447$, $p < .0001$). Hukey-HSD post-hoc

---

[1]As the *RR-scores* are normalized by subtracting the mean *RR-score* of the random baseline, the random baseline strategy itself has a normalized *RR-score* of 0.

comparisons indicated that the strategies discovered by MGPS are significantly more resource-rational than the strategies discovered by PO-UCT with 10 steps ($p < .0001$, $d = 2.18$), 100 steps ($p < .0001$, $d = .27$), 1000 steps ($p < .0001$, $d = .14$), or 5000 steps ($p < .0001$, $d = .11$). While MGPS achieves significantly higher *RR-scores* than all PO-UCT variants, the size of the effect decreases from a very large effect to a small effect when increasing PO-UCT's computational budget sufficiently. We therefore expect that PO-UCT with a much more than 5000 steps would ultimately achieve comparable *RR-scores* to MGPO, albeit at a much higher computational cost. Moreover, MGPS was substantially faster than PO-UCT with a computational budget of 1000 steps or more, which is when PO-UCT's performance starts to approach that of MGPS. With a computational budget of 100 steps or fewer, PO-UCT is faster than MGPS. However, such a small computational budget is not enough for PO-UCT to discover strategies with a *RR-score* anywhere near that of the strategy discovered by MGPS. Critically, the high amount of computation required for PO-UCT to achieve an approximately similar level of resource-rationality would render PO-UCT unusable for a cognitive tutor that computes feedback in real time.

## 7 Conclusion

People's decision-making is prone to systematic errors [16], and although people are happy to delegate some decisions, most CEOs are unlikely to let AI decide which projects their company should work on. Moreover, people are reluctant to use the more accurate technical decision procedures because they tend to be more tedious [28, 21, 18]. Motivated by people's insistence on making their own decisions with simple heuristics, we leveraged AI to discover and teach decision strategies that perform substantially better than people's intuitive strategies but are nevertheless simple enough that people use them. To this end, we introduced a metareasoning method for leveraging AI to discover optimal decision strategies for human project selection. Modeling project selection through the lens of resource rationality allowed is to formulate a mathematically precise criterion for the quality of decision strategies for human project selection. We further develop an efficient automatic strategy discovery algorithm automatically discovers efficient strategies for human project selection. Our algorithm discovered decision strategies that are much more resource-rational than the strategies humans discovered on their own and the strategies discovered by a general-purpose algorithm for solving POMDPs (PO-UCT). Using the efficient decision strategies discovered by our algorithm, we create a cognitive tutor that uses a shaping schedule and metacognitive feedback to teach the strategies to humans. In the training experiment, our cognitive tutor fostered significant improvements in participants' resource rationality.

A main limitation of our method is that it is unknown how precisely the environment parameters need to be estimated to construct the metareasoning task, which can prove especially problematic when there isn't much data on past decisions. Future work could investigate and potentially address it by extending MGPS with a Bayesian inference approach to estimate the environment structure. Encouraged by the promising results from successfully teaching humans in our naturalistic model of project selection, we are excited about future work assessing the real-world impact of improving people's decision-making by evaluating their decisions directly in the real world. Additionally, we are also excited about potential future work that combines MGPS with AI-Interpret [32] to automatically derive human-legible recommendations for how to make project selection decisions. Lastly, although MGPS performed very well on our benchmarks, MGPS's myopic approximation could fail in more complicated scenarios with interdependent criteria. Such challenges could be addressed by solving meta-level MDPs with methods from deep reinforcement learning, for example by utilizing AlphaZero [30].

Our results indicate that it is possible to use resource-rational analysis combined with automatic strategy discovery to improve human decision-making in a realistic project selection problem. As selecting projects is a common problem faced by both organizations and individuals, improving their decision strategies in this setting would have a direct positive impact. For example, a project-selection tutor could be integrated into MBA programs to teach future decision-makers efficient decision strategies as part of their education. We are optimistic that our general methodology is also applicable to other real-world problems, offering a promising pathway to teach people efficient strategies for making better decisions in other areas as well. Besides project selection problems, we believe our approach could be used to improve real-world decision-making in areas such as career choice, grant making, and public policy.

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
