# OpenReview forum: "Leveraging automatic strategy discovery to teach people how to select better projects"
_NeurIPS.cc/2023/Conference — Submitted to NeurIPS 2023_

### Official Review · Reviewer_SQT4 · 2023-06-30

**Soundness:** 4 excellent
**Presentation:** 4 excellent
**Contribution:** 3 good
**Rating:** 5
**Confidence:** 4

**Summary:**

This works addresses two problems: first, the problem of solving a specific class of information-gathering decision processes, and second, applying the resulting algorithm in the real world by creating an intelligent tutoring system. The algorithm is a straightforward greedy optimizer that gathers information based on maximizing one-step value of information.  The intelligent tutor blends the algorithm with a UI that gives humans feedback on their decisions by comparing them to the algorithm's; the paper shows that the humans implicitly learn from this feedback and improve their decision making abilities.  The paper concludes with a few small comparisons of the algorithm to other algorithms.

**Strengths:**

+ The paper is well-written
+ The problem seems reasonably well formulated
+ The algorithm is natural and reasonable
+ The algorithm seems to perform well compared to alternatives
+ The "intelligent tutor" is a bit of a misnomer, but does seem to improve human decision making ability

**Weaknesses:**

- This paper blends two distinct ideas, and does not take either very deep. The MGPS algorithm is straightforward, and the authors spend very little time (for example) analyzing its properties or connecting it seriously with the vast literature on decision processes.  (For example, I'm surprised not to see any connection to bandits).  I think you could have written a whole paper about MGPS -- establishing properties, contrasting with other algorithms, etc.  For example, a regret bound or similar form of theoretical analysis would have been nice.

On the other hand, the "intelligent tutor" seems like a very simple UI that gives very basic feedback.  While the authors do demonstrate some effectiveness, it too is not investigated deeply.  There is no comparison, for example, to other forms of UI, to other forms of feedback, etc.

So, it's hard to say: is this paper about MGPS, or intelligent tutor UI/UX design?  And while I appreciate that the authors probably have a vision of "solving a real-world problem" with this combination of ideas, I have to say that the problem formulation seems pretty far away from something that could actually be used in a business decision support system.


**Questions:**

1) I am surprised by the findings of the paper.  While the authors repeatedly refer to MGPS as generating "optimal strategies" and humans as using "simple heuristics", this terminology is hard to justify, since MGPS is itself a greedy heuristic.  I would have intuitively thought that humans would also try to maximize one-step information gain.  My question is this: do the authors have any sense of what strategy humans were using before exposure to the tutor? Why is their intuitive heuristic so bad? Can they (or you) somehow articulate what the humans are learning?

**Limitations:**

The authors do not explicitly discuss the limitations of their algorithm/UI in a dedicated section.

I do not see any potential negative societal impact.

---

> ### Author Rebuttal · Authors · 2023-08-09
>
> We agree that the strategies discovered by MGPS are not “optimal strategies” as MGPS makes use of a greedy heuristic and will reword the manuscript accordingly.
>
> MGPS identifies a near-optimal greedy planning heuristic by approximating the value of computation for each available planning action. It is plausible that humans also try to maximize one-step information gain, but their ability to do so is very limited because they don’t know the probabilistic structure of the environment. Consequently, to the extent that people attempt to maximize the one-step information gain, they do so in a highly suboptimal way:
> 1) Looking at the click agreement measure, humans don’t seem to learn the strategy taught by our tutor on their own (as click agreement for the “no feedback” condition is lower than for the “choice tutor” condition).
> 2) We analyzed in which percentage of test trials participants investigated the most important criteria (criteria 5, “organizational readiness) first, an important component of the near-optimal strategy discovered by MGPS. In the “MGPS tutor” condition, participants investigate the criteria as their first planning action in 67% of test trials, while in the “no tutor” control condition, participants only do so in 44% of the test trials.
> 3) Similarly, we also investigated if participants query the most reliable experts (experts 2 and 6) first, another component of MGPS’ strategy. In the “MGPS tutor” condition, participants do this in 76% of test trials, while participants in the “no tutor” condition only achieve query the most reliable experts in 55% of test trials.
> 4) These results indicate that participants not taught by the MGPS tutor fail to prioritize the most important criteria and the most reliable experts very early, only selecting the “correct” first planning action (i.e. the correct criteria and experts) in 33% of test cases.
> 5) Moreover, participants assigned to the “no feedback” condition performed slightly less planning operations on average (3.6 planning operations compared to 3.9 for the “dummy tutor” and 4 for the “choice tutor” condition). This indicates they learned a worse strategy for deciding when to stop planning.
> Prior work on strategy discovery produced similar results, where humans performed worse than one-step planning (i.e. [14]).
>
> Additionally, prior work on strategy discovery produced similar results, where humans performed worse than one-step planning (i.e. [14]).

---

### Official Review · Reviewer_3k1i · 2023-07-05

**Soundness:** 2 fair
**Presentation:** 2 fair
**Contribution:** 1 poor
**Rating:** 4
**Confidence:** 3

**Summary:**

The problem this paper tries to tackle is how to improve human decision making in the specific problem of selecting a project between a candidate set of existing projects. The strategy to improve the human's decision making is to build an agent that can solve the project selection itself by framing as a POMDP and then having that agent serve as a tutor during a teaching phase. The agent is learned by first framing the problems as a POMDP with Guassian states and the paper proposes a myopic algorithm to select actions (actions allow you to acquire information about each project on a specific dimension of the Guassian). The main evaluation of the paper is with a user study where human participants are trying to select between multiple projects in terms of estimated performance. The authors show that participants who received training with the tutor performed better than participants without a tutor or with a dummy tutor.

**Strengths:**

- very well-designed and rigorous user study that shows that the agent and the tutoring was effective.


**Weaknesses:**

- (main reason for low rating) out of scope of NeurIPS: this paper seems like a bad fit for NeurIPS, the paper does not contribute new algorithms that are broadly applicable and does not evaluate methods on broadly recognized datasets or benchmarks.  This papers models the problem of human project selection as a PODMP and proposes a relatively straightforward procedure and evaluates on a synthetic task of project selection. I don't see any insights that the community might benefit from. I think this would be a strong paper for a conference that suits it more.
A bad (but still useful) heuristic is to look at the references where I count a single NeurIPS paper (from 2010) and one UAI paper (among a lot of management and behavioral science citations that don't include other human-ai conferences like CHI).

- why not display the recommendation of the AI agent during test time as a baseline? since the AI agent performs well at the task, participants should be able to see its recommendations at test time.

- the proposed algorithm while adequate for the problem is not a generalizable solution for the broad family of problems that are of interest in the human-AI space. In particular, the myopic approximation is very limiting. While section 6 does evaluate it against (a relatively old) baseline in PO-UCT, it limits the baseline to 5000 steps because of runtime constraints, however, it might be possible to optimize the performance of PO-UCT to be faster.

- novelty: the method is a modification over MGPO [14] which introduces the myopic approximation, the proposed method MGPS modifies it to make it suitable for project selection. The authors do a good job of comparing to MGPO, but novelty overall is limited.

- value of tutoring for project selection: why not just follow the recommendations of MGPS? the tutoring is specific to the problem domain, thus I don't understand anything that the human can take away from the tutoring to future tasks except the one they will encounter. Moreover, for the real world problem presented, it is a one shot decision problem, so tutoring is not as well motivated.

**Questions:**

(from weaknesses) why not display the recommendation of the AI agent during test time as a baseline? since the AI agent performs well at the task, participants should be able see it's recommendations at test time.

for section 6, how are the different environments of the project selection task generated?

Comments:
- in title and sections, first letter of each word must be in capital


**Limitations:**

limitations are well discussed.

---

> ### Author Rebuttal · Authors · 2023-08-09
>
> Question 1: Our intention behind the human training experiment was to teach people how to use the decision strategy discovered by MGPS themselves, as opposed to replacing the human decision maker or providing a tool that is to be used in an online fashion. Showing the recommended action during test time would therefore fail to evaluate to what extent the participant learned to apply the decision strategy, as we would expect participants to simply follow the recommendations instead of planning for themselves.
>
> Question 2: Apologies for not including this in the main manuscript, we will add it in the revision process. An environment instance is generated by (1) sampling ground truth rewards for each project’s criteria from the criteria’s reward distribution (which is the same distribution as the initial belief state), and (2) randomly sampling the expert’s guesses based on the expert’s reliability (precision parameter) and the criteria’s ground truth value.

---

> > ### Comment · Reviewer_3k1i · 2023-08-16
> > **Response**
> >
> > Thank you for your rebuttal. I have raised my score to 4, the reason for the low score is that I don't think the approach proposed has enough novelty over prior work and that the user study evaluation is not of general interest.

---

### Official Review · Reviewer_RmKP · 2023-07-06

**Soundness:** 3 good
**Presentation:** 3 good
**Contribution:** 2 fair
**Rating:** 3
**Confidence:** 4

**Summary:**

This paper focuses on the problem of project selection (how does a human choose which, among a set of possible projects, is the best one to pursue). To address this problem, they develop an algorithm called MGPS that discovers a rational greedy strategy for solving this problem, and then they attempt to teach this strategy to a human via an intelligent tutoring system. The approach is evaluated in a real world project selection scenario.

- page 2, line 64: "selection.To" -> "selection. To"
- page 3, line 127: "we introduce explain our general" -> "we introduce our general"


**Strengths:**

It is interesting to see a paper that goes all the way from finding an algorithm to solve a problem, to teaching humans based on that strategy and then evaluating human performance after being taught.


**Weaknesses:**

The same strength of the paper seems to also be its main weakness: because the authors try to tackle a whole large problem end to end, I did not find each if the individual pieces that ground breaking. Perhaps if the paper focused on just one of the problems (finding optimal strategies, or just better techniques for teaching the learned strategies), maybe a stronger contribution would arise, by more systematically addressing one problem. But as it stands, the paper seems two-headed, with limited contributions, as each of the two problems is only dealt with shallowly.


**Questions:**

- Why limit yourself to a greedy approach? Given Algorithm 1, it is trivial to define a version that does a fixed amount of look-ahead (say, k steps) to find the optimal action considering the next k actions, rather than just the next (k=)1, as the current one does.
- How come PO-UCT took so long to run for just 5000 steps? That is ~500 steps per second, which is extremely slow for what I would expect. Was a standard implementation used?
- It also surprises me that PO-UCT underperforms MGPS, is there is something I do not understand. If PO-UCT is using the UCB bandit strategy, and there are, say, 500 possible actions at the root node (which are the 500 actions that MGPS will consider, and then select the best), during the first 500 iterations, PO-UCT will systematically select each of those 500, until each of them is expanded at least once, before moving on to the exploration/exploitation phase of UCB. Hence, at this point, shouldn't it be equivalent to MGPS? (they both have explored all actions), and then PO-UCT become better and better after this point? What am I missing?


**Limitations:**

See weaknesses.

---

> ### Author Rebuttal · Authors · 2023-08-09
>
> Question 1:
> - We briefly experimented with a multistep version of MGPS when designing the algorithm. For numbers of steps larger than 1, the computational complexity increases rapidly, as it requires discretizing the belief state updates and searching through an exponentially increasing state space. Additionally, we did not observe a large increase in performance from doing this, indicating that greedy strategies are sufficiently resource-rational in our environment distribution.
> - Our main motivation behind limiting MGPS to a single step and evaluating runtimes in general is that our intelligent tutor requires MGPS to run while the learner interacts with it. This requires the computation to be carried out online in an efficient and scalable manner. Precomputing optimal actions is impossible due to the exponential nature of possible belief states.
>
> Question 2 and 3:
> - The main limitation of PO-UCT is that it is exploring the state of belief states, where sampling an action only reveals a noisy observation based on the current belief state. MPGS, on the other hand, does not rely on sampling as it calculates the expected value of computation directly.
> - Our project selection environment does not directly fit into existing POMDP frameworks, which led us to implement PO-UCT ourselves. There is likely to be room for optimization in both our MGPS and PO-UCT implementation. The 500 steps per second are a result of the fact that each step requires to compute the posterior belief state using Bayesian inference.

---

### Official Review · Reviewer_teWD · 2023-07-06

**Soundness:** 4 excellent
**Presentation:** 4 excellent
**Contribution:** 2 fair
**Rating:** 3
**Confidence:** 4

**Summary:**

The authors pose the problem of teaching decision-makers how to take a single action (picking a project from among a set of projects) based upon costly advice from experts across different, weighted criteria. The authors develop a reinforcement learning approach to creating a tutor that approximately solves the MDP (using a myopic approximation). The proposed tutor outperforms a baseline on a banking dataset. The tutor is also shown to improve decision-making of subjects in an online study, who get to learn from the tutor by watching the tutor make decisions.

**Strengths:**

+Section 3 is very clearly written. Well done (though Line 162 could have used O-notation).
+The paper clearly presents the algorithm.
+The paper is addressing an important problem of developing a tutoring system for solving MDPs
+The paper has strong empirical results vs. PO-UCT.
+The paper shows statistically significant results in a user-study. It is nice that a user-study was done.

**Weaknesses:**

-The terms h, e, \lambda, N, and R_{total} are not sufficiently defined in Lines 41-50. Perhaps it would be better to give a more complete description later in the paper and abstract the presentation here to make it more intuitive just with words.
-The comment on brainstorming in Line 86 ignores relevant literature on the wisdom of the crowd, the science behind brainstorming and focus groups, etc.
-The statistical analysis does not report testing for the meeting of a Gaussian assumption for the confidence intervals. Details of the Box approximation should be provided. Further, it would have been better to also report p-values in that table.
-For the user study, Table 1 should report how optimal (the RR) the MGPS tutor would be if run automatically (no human intervention) and how poorly a random, automatic system would be for the RR-score.
-I am uncertain that the authors are reporting the degrees of the freedom of the F-test. The F-test has two degrees of freedom, but only 1 seems to be indicated in Lines 295-299.
-I can appreciate that the authors might have thought that the results in lines 329-345 indicate that MGPS > PO-UCT and therefore not included PO-UCT as a baseline in the user study; however, that is a debatable decision. It could be that the behavior of PO-UCT is more intelligible by users, and users with PO-UCT could have outperformed those trained by MGPS. As such, I recommend the user study be re-ran (to account for cohort effects) with the PO-UCT baseline and randomizing the allocation of participants to the conditions.
-The paper isn't exactly "tutoring" participants. Rather, the system is providing recommendations (or making decisions) and the users have to reverse engineering the actual strategy. Considering that the strategy itself is not constrained to be a set of if-then rules, it is unclear what exactly is being learned or how. It would make the paper better to have actually analyzed (by collecting the data from users) what users are learning and thinking. I would recommend looking at methods in explainable Artificial Intelligence.

Note:
-I recommend the "Dummy" tutor be called a random tutor -- a "random tutor" is a more clear description of that condition.

**Questions:**

-Arguably, the object-level decision is not "really" an MDP as there is no sequential decision-making. Could the authors comment on why the MDP formulation was adopted for this formulation rather than a multi-arm bandit formulation?


**Limitations:**

Limitations are reasonably discussed.

---

> ### Author Rebuttal · Authors · 2023-08-09
>
> Question 1: In this case, a multi-armed bandit formulation would be sufficient to model the object-level decision. As the multi-armed bandit problem is a special case of the more general MDP, we chose the MDP formulation to keep our environment model as general as possible. MGPS does not require a one-step object-level task, and this formulation allows the approach to be used on sequential project selection problems in the future.
>
> Thank you for pointing out a number of smaller errors in our manuscript, we will update the article accordingly. Regarding using PO-UCT as a baseline for the user study: one of the advantages of MGPS is that it can be run online in our tutor to compute feedback depending on the learner’s current belief state. It is not possible to precompute this feedback, as the space of possible belief states is exponentially explosive. MGPS runs significantly slower, which makes it much less practical for use in online tutoring sessions.

---

> > ### Comment · Reviewer_teWD · 2023-08-14
> > **Response by Reviewer**
> >
> > Thanks to the reviewers for answering one of my questions and acknowledging the writing issues. I also appreciate the discussion regarding PO-UCT. However, it is unclear why it is "not possible" to approximately compute feedback. It would seem a variety of online multi-armed bandit approaches could work as well.
> >
> > I look forward to the reviewers discussing additional points of feedback from my original review.

---

> > > ### Author Response · Authors · 2023-08-21
> > >
> > > Thank you for your additional feedback.
> > >
> > > The statistical test we are using is nonparametric and does not require the data to follow a Gaussian distribution. The confidence intervals provided in the table are not part of the statistical analysis and are only meant to give the reader an idea of the range of scores. The reported statistic has only one degree of freedom because the denominator degree of freedom is set to infinity. Further details about the used statistical test can be found in the documentation of the nparLD package (https://search.r-project.org/CRAN/refmans/nparLD/html/f1.ld.f1.html).
> > >
> > > The tutor is tutoring participants in the sense that it is letting users make choices between different actions and then provides feedback based on the chosen action’s resource-rationality. The recommendations are simplifying the problem by isolating specific aspects of the project selection task (e.g. learning which expert to query) but the main learning mechanism is the provided feedback.
> > >
> > > Thank you for pointing out the missing link to literature on the wisdom of the crowds and the unclear description of some of the parameters. We will add these and the requested measures regarding the human experiment to Table 1 in the revision process (both p-values and the RR-score of MGPS are already given in the manuscript text). We don’t report the RR-score of a random baseline, as we already normalized the reported scores against it (i.e. its score would be 0).

---

### Official Review · Reviewer_uoEm · 2023-07-08

**Soundness:** 3 good
**Presentation:** 3 good
**Contribution:** 2 fair
**Rating:** 5
**Confidence:** 5

**Summary:**

This paper addresses the problems of (1) sequential decision-making of information gathering through asking experts for information about the rewards for different projects (as meta-reasoning towards project selection), and (2) teaching people how to make near optimal decisions in the same problem through training in an intelligent tutoring system.  Specifically, novel problem aspects are considered compared to recent work on similar problems, including the availability of different experts of different reliabilities (instead of one information source) and multiple criteria for evaluating the quality/reward of a project.  An algorithm is developed for generating solutions guiding information gathering.  That solution is used to guide the training of humans through an Intelligent Tutoring System (ITS).  Experimental results highlight the benefits of the approach through both better training of humans in an ITS (measured through better rewards earned) and better performance in simulation of agent reasoning than the problem modeled as a belief MDP solved by PO-UCT.

**Strengths:**

The primary strengths of this paper include:

S1) The problem addressed is one that is relevant in the human-AI applications and it considers novelties not previously addressed that are important real-world complexities.  The problem of sequential decision-making will be of interest to the planning community at NeurIPs.

S2) The paper is well written and easy to follow.

S3) Modeling the problem as a POMDP is appropriate, and the MCDM component seems to be appropriately modeled.

S4) I appreciated the use of two very different sets of experiments -- both training humans within an ITS and simulating the approach directly.  The ITS experiment was well designed and the evaluation was very carefully conducted and convincing.

**Weaknesses:**

The primary weaknesses of this paper include:

W1) The main contributions identified in the abstract are incremental, seemingly adding some environment complexity and solution adaption to [14], rather than being entirely novel.

W2) While the choice of a POMDP was appropriate, I wasn't entirely sure why the authors relied more on the belief MDP formulation rather than a true POMDP with observations separate from belief state transitions.

The problem being solved is gathering information to ultimately choose the best task to accomplish.  This is very closely related to prior POMDP usage for problems like preference elicitation where the agent gathers information about which is the user's main preference or task they want the agent to perform.

Boutilier,C. 2002.A POMDP Formulation of Preference Elicitation Problems.In Proceedings of AAAI'02, pp. 239–246.

Doshi, F., & Roy, N. 2008. The Permutable POMDP: Fast Solutions to POMDPs for Preference Elicitation. In Proceedings of AAMAS'08, pp. 493–500.

In those models, the state space is the set of possible tasks/preferences, and actions either (1) query an information source (e.g., the user) for observations used to update the agent's Bayesian beliefs about the top preference/task, or (2) end information gathering to perform the perference/task the agent thinks is the top.  That is fundamentally the same as the problem being addressed here, but the details are slightly different.  Instead, your state space is belief states over the details of the tasks, from which a top one is selected.  It's not clear to me why the former wouldn't work in this situation and what the advantage is in your formulation, which would help strengthen the novelty of the work. (Note: what makes your paper different from [14] also makes it different from those works).

Information gathering in POMDPs in general also have special formulations, such as the \rho-POMDP and equivalent POMDP-IR, and I would think your problem model would also fit nicely in the POMDP-IR, but neither is considered in your related work.

\rho-POMDP: Araya-Lopez, M., Buffet, O., Thomas, V., & Charpillet, F. 2010. A POMDP Extension with Belief-Dependent Rewards. In Proceedings of NIPS'10.

POMDP-IR: Spaan, M.T.J., Veiga, T.S., & Lima, P.U. Decision-theoretic planning under uncertainty with information rewards for active cooperative perception. Journal of Autonomous Agents and Multiagent Systems, 29(6):1157-1185.

W3) I also wasn't sure why you chose PO-UCT as your baseline?  Its the simpler version of POMCP (whereas POMCP would have been more appropriate if you had explicit observations in your model).  Belief MDPs can also be considered continuous state MDPs, and there have been many advancements in Monte Carlo Tree Search planning in that area since PO-UCT, such as:

Sunberg, Z. & Kochenderfer, M. 2017. Online algorithms for POMDPs with continuous state, action, and observation spaces. arXiv, 2017. doi: 10.48550/ ARXIV.1709.06196. URL https://arxiv.org/ abs/1709.06196.

Finally, the \rho-POMDP has a MCTS solution that would also be relevant that is more recent:

Thomas, V., Hutin, G., & Buffet, O. 2020.. Monte information-oriented planning. In Proceedings ECAI’20.

W4) I also didn't quite understand the novelty of the ITS experiment compared with [14].  Was the only difference the change in the meta-reasoning algorithm, or were there other differences?

**Questions:**

Q1) Why did you use your specific formulation of the POMDP problem instead of the state-of-the-art \rho-POMDP or POMDP-IR for guiding information gathering?

Q2) Why did you choose PO-UCT as your baseline?

### Post-Rebuttal ###

I thank the authors' for their rebuttal and the ensuing conversation.  They helped me better understand the research and its place in the literature.

**Limitations:**

These were appropriate addressed.

---

> ### Author Rebuttal · Authors · 2023-08-09
>
> Question 1:
> - Our work is grounded in prior work on metareasoning and strategy discovery which has been modeled with metalevel-MDPs in the past, making it the natural candidate for our adapted problem setting (e.g. [4], [6], [10], [14]).
> - We chose the POMDP framework because it was sufficient to model all relevant aspects of the project selection problem, and we didn’t see the need to adopt a more complex framework. Both of the suggested POMDP variants are aimed at settings where object-level rewards are inaccessible to evaluate agents, and the agent is instead rewarded for information gathering directly. In our setting, the object-level reward signal is given through the selection of a project which does not depend on the agent’s belief state. The advantage of our POMDP formulation is that it doesn’t require the design of additional information rewards or information communication (“commit”) actions. It is plausible that our approach could be reframed in an adapted \rho-MDP by creating a reward function that rewards the value of information gain similar as it is computed in MGPS. However, this would require to externalize a large component of our strategy discover algorithm into the environment, making it unclear how to learn efficiently in this domain and making comparisons between algorithms more difficult.
>
> Question 2:
> POMCP combines PO-UCT with Monte-Carlo belief state updates, in which the belief state is approximated with particle filters. As our environment allows us to compute the belief state updates directly, we were able to apply PO-UCT to the problem directly without requiring an approximation to the belief state.

---

> > ### Comment · Reviewer_uoEm · 2023-08-18
> > **RE: Rebuttal by Authors**
> >
> > I thank the authors for their rebuttal and for the clarifications to all reviewers that answered some of my questions buried elsewhere in the review.  I especially better understand how the user-study is novel over [14].
> >
> > I think your answer to Q2 makes perfect sense for why you'd use PO-UCT as the baseline over POMCP.  I was also wondering if there was a reason you didn't consider more recent MCTS algorithms for POMDPs (e.g., DESPOT, HyP-DESPOT, \alpha-DESPOT, PUCT, etc.).  While there are still plenty of advances coming from extending POMCP, so it's not a bad place to start, POMCP is also not the state-of-the-art, and I'm unsure whether a more recent algorithm might have done better with a limited computational budget, especially since PO-UCT seems to be converging to values close to your algorithm.
> >
> > With regards to Q1, I would think that the POMDP-IR wouldn't have a problem modeling object-level rewards since its reward function is a combination of the standard R(s, a) reward model plus information rewards (which is one of the reasons people choose to use this model over the equivalent \rho-POMDP).  Modeling the rewards for selecting a project would be done through R(s, a), where the action space would include an action for choosing each project (and that action would have 0 information reward, whereas interacting with the experts would have their own actions that have information rewards but no object-level rewards).  Such a decision process might not outperform your approach, but it seems like a natural baseline that would help the reader better evaluate your model, especially since even the simpler POMDP with older POMCP seems to be converging to the same performance.
> >
> > While I see how your work is novel in your problem setting compared to [4, 5, 10, 14], I'm still not sure how it is novel compared to other uses of information gathering with POMDPs.  The Doshi and Roy (2008) paper I mentioned in my review also uses a POMDP to guide metalevel information gathering (to ascertain a user's preference) before receiving an object-level reward for acting on it, it considers information sources of different quality (two ways of gathering information from the user, equivalent to asking different noisy experts), and has a limited budget for information gathering before acting, which are several of the novelties you've highlighted in your rebuttal over past problems in your particular problem setting.  Can you further clarify?

---

> > > ### Author Response · Authors · 2023-08-21
> > >
> > > Thank you for your further feedback and pointing out how we can incorporate past work as additional baselines. We agree that the problem could be rephrased in such a way, although we still believe that manually designing useful information rewards would externalize an important component of the strategy that MGPS discovers automatically in the full metalevel-MDP formulation.
> > >
> > > The main difference we see to Doshi and Roy (2008) is that the metalevel-MDP formulation does not assume symmetry in the model (e.g. some properties of specific projects could be harder to evaluate than other projects) and therefore solves a more general problem as the permutable POMDP. For example, the metalevel-MDP structure used in [10] makes it necessary to evaluate specific goals first, as they are drawn from a wider distribution of rewards. Due to the wide range of possible belief states in the project selection task, we are also skeptical that the proposed solution algorithm based on value iteration would be feasible to run with our tutoring system, as we require the value of information gathering actions to be efficiently computed online.

---

### Author Rebuttal · Authors · 2023-08-10

Response to concerns about fit and novelty

Since multiple reviewers expressed similar concerns regarding the novelty of MGPS and the intelligent tutor, we will address these in a single response. The novelty of our work lies in (1) the development of a new strategy discovery algorithm (MGPS), (2) formalizing the project selection problem, (3) the development of a new intelligent tutor that can improve human decision-making on the project selection task. Below, we briefly elaborate on the significance and substance of these three innovations:
1. Novelty of MGPS: MGPS contains several technical advancements that extend the scope of strategy discovery methods to the project selection setting. The main advances compared to the previous state-of-the-art strategy discovery method [14] are: MGPS estimates the expected value of computation for discrete expert guesses, selects computations between multiple experts with varying reliabilities, plans within a fixed budget, and evaluates a project based on multiple criteria of different importance.
2. Novelty of the problem/application: We provided the first formal model of a practically important problem, namely discovering optimal strategies for project selection under time and information constraints, and developed a benchmark for it. Unlike previous formulations of the project selection problem, our formulation captures that the decision has to be made within a limited time and that this makes it impossible to gather all relevant information. To achieve this, we modelled the task of project selection as a metalevel-MDP. This constitutes the first metalevel-MDP model of a real-world task and estimates important parameters (e.g. expert reliability, project outcomes, criteria importance) from real data. This significantly increases the real-world relevance of this line of research by advancing it from highly artificial toy problems toward realistic decision problems that organizations face in the real world. We believe our work to be a stepping stone to improving human decision-making in relevant real-world tasks.
3. Novelty of the MGPS Tutor: Our intelligent tutor used in the human experiment differed from the one introduced in [14] in multiple ways: the tutor teaches participants planning strategies in the project selection task, a more realistic application than the scenarios used in prior work that added multiple new features (e.g. planning limitations, choosing between multiple experts with different reliabilities). To teach strategies in the project selection task, the tutor relies on MGPS to compute the approximate value of computation of meta-level actions and features a new shaping schedule varying the type of selection (between projects, experts, and criteria).
We believe these advancements make our work suitable for NeurIPS.

Moreover, we believe that our submission falls well within the broad scope of NeurIPS. This is evident from the fact that NeurIPS has previously published several articles on improving human decision-making, including “Reliable Decision Support using Counterfactual Models” and “Closing the loop in medical decision support by understanding clinical decision-making: A case study on organ transplantation”, numerous applications of AI to teaching people skills and knowledge, including  “Assistive Teaching of Motor Control Tasks to Humans”, “Understanding the Role of Adaptivity in Machine Teaching: The Case of Version Space Learners”, “Machine Teaching of Active Sequential Learners”, “Optimal Teaching for Limited-Capacity Human Learners”, “Curriculum Design for Teaching via Demonstrations: Theory and Application”, “Learner-aware Teaching: Inverse Reinforcement Learning with Preferences and Constraints”, “Learning to Teach with Dynamic Loss Functions”  “Automatic Discovery of Cognitive Skills to Improve the Prediction of Student Learning”, as well as cognitive science research on human teaching, including “Showing versus doing: Teaching by demonstration” and “How Do Humans Teach: On Curriculum Learning and Teaching Dimension”.

---

### Decision · Program_Chairs · 2023-09-21

**Decision:**

Reject

**Comment:**

The paper studies a sequential decision-making problem that requires information gathering to select a project to pursue. An algorithm is proposed to find an approximate strategy for solving this problem, and then an intelligent tutoring system is developed to teach this strategy to humans. The reviewers acknowledged that the paper considers an important human-AI application setting and appreciated the effort in conducting studies with human participants. However, the reviewers pointed out several weaknesses in the paper, and shared common concerns about limited novelty and lack of technical depth in different components of the methodology. We want to thank the authors for their detailed responses. Based on the reviewers’ concerns and follow-up discussions, unfortunately, the final decision is a rejection. The reviewers have provided detailed and constructive feedback to the authors. We hope that the authors can incorporate this feedback when preparing future revisions of the paper.